# Prevalence and determinants of maternal psychological distress: A hospital- based cross-sectional study among pregnant women attending antenatal care in Dar-es-salaam, Tanzania

Glory Cuthbert[1,2]*, Samuel Likindikoki[2], Deodatus Kakoko[3], Muzdalifat Abeid[4]

**1** Kilimanjaro Fertility Institute, Arusha, Tanzania, **2** Department of Psychiatry and Mental Health, Muhimbili University of Health and Allied Sciences, Dar-es-salaam, Tanzania, **3** Department of Behavioral Sciences, Muhimbili University of Health and Allied Sciences, Dar-es-salaam, Tanzania, **4** Department of Obstetrics & Gynecology, Agha-Khan Hospital, Dar-es-salaam, Tanzania

\* gcathode@yahoo.com

## Abstract

Pregnancy is a period of great transition in women, marked by various physiological, emotional, and social adjustment that can increase a woman's vulnerability to psychological distress during this critical period. Despite its potential impact on maternal and fetal health, maternal psychological distress remains underexplored in Tanzania, with limited published data on its burden. This study aimed to assess the prevalence and associated factors of maternal psychological distress among pregnant women receiving antenatal care at the Muhimbili National Hospital in Dar-es-Salaam. A hospital-based cross-sectional study was conducted with 287 pregnant women attending antenatal care (ANC) at Muhimbili National Hospital. A probability sampling method was used to select study participants via a simple random sampling technique through a lottery method. The Oslo Social Support Scale (OSSS-3) was employed to assess perceived social support, while the WHO Multi-Country Study on Women's Health and Domestic Violence against Women tool was used to measure intimate partner violence (IPV). Maternal psychological distress was assessed using the Kessler Psychological Distress Scale (K10). Data were analyzed using Poisson regression with robust variance to estimate crude and adjusted prevalence ratios to determine factors associated with maternal psychological distress, with statistical significance set at $p < 0.05$ and a 95% confidence interval. The mean age of participants was 30.87 (SD ± 4.3). The prevalence of maternal psychological distress was found to be 32.1%. Pregnant women aged 18–24 years had a significantly higher prevalence of psychological distress compared to those aged 25 years and above (aPR = 1.64, 95% CI: 1.10–2.45, p = 0.016).Women with perceived poor social support (aPR = 0.458, 95% CI: 0.321–0.652, p < 0.001) and moderate social support

which permits unrestricted use, distribution, and reproduction in any medium, provided the original author and source are credited.

**Data availability statement:** All relevant data have been attached as Supporting Information.

**Funding:** The authors received no specific funding for this work.

**Competing interests:** The authors have declared that no competing interests exist.

(aPR = 0.325, 95% CI: 0.178–0.593, p < 0.001) were also strongly associated with higher prevalence of psychological distress compared to those with good support. The findings revealed that maternal psychological distress is prevalent among pregnant women in this setting, highlighting the need to integrate early mental health screening and psychosocial support into ANC services.

## Introduction

Emotional changes are natural part of the pregnancy, but sometimes they can become overwhelming, affecting a woman's mental wellbeing and leading to psychological distress. Psychological distress broadly encompasses non-specific emotional difficulties, such as feelings of stress, anxiety, low mood, or general discomfort that can disrupt daily life [1]. For many women, pregnancy can be a period of heightened emotional vulnerability with distress symptoms, potentially impacting day to day functioning, family relationships, and self-care [2]. Beyond these personal struggles, research indicates that high levels of psychological distress can negatively impact fetal growth and development, increasing the risk of cognitive, behavioral, and physical challenges in childhood, and future mother-infant bonding [3,4].

Globally, it has been reported that 15% to 20% of women of reproductive age experience psychological distress, including symptoms of anxiety and depression, at any given time [5] On the other hand, global prevalence of psychological distress among women during pregnancy has been reported in the range from 10% to 25% [6,7], indicating a heightened vulnerability during this critical period. The experience of psychological distress during pregnancy was described as arising from an imbalance between social-environmental demands and an individual's coping resources. Factors such as the need for role adaptation, shifts in identity, and perceived loss of control are mentioned to further contribute to distress [8]. High distress levels are considered as early indicators of perinatal mental health disorders, such as anxiety and depression, which can persist into postpartum and negatively affect long-term maternal and child health [9].

Several risk factors have been reported to contribute to psychological distress. This includes poor social support, younger maternal age, low educational attainment, history of obstetric complications, and exposure to intimate partner violence (IPV) [7]. In Tanzania mental health morbidity during pregnancy was reported to be associated with IPV, inadequate family support and being unmarried [10]. Similarly in a qualitative study which was done in Dar es Salaam, found that, pregnant women reported distress emanating from poverty, partner conflict and limited autonomy in decision making [11]. Studies from neighboring East African countries particularly Kenya and Uganda respectively, have also highlighted similar patterns, and heightened level of psychological distress during pregnancy as a contributor to various mental health challenges during pregnancy [12,13]. Despite of these reported evidence, a comprehensive epidemiological data on prevalence and socio-cultural determinants of psychological distress among pregnant women in Tanzania, is still limited by the

scarcity of published information. To address this gap, this study aimed to assess the prevalence and associated factors of maternal psychological distress among pregnant women attending antenatal care at Muhimbili National Hospital in Dar-es-Salaam, Tanzania to inform the burden and promote early interventions within antenatal care (ANC) settings.

## Methods

### Study design and settings

This was a hospital-based cross-sectional study, conducted from 4th March to 31st May 2024 at Muhimbili National Hospital (MNH), a tertiary referral and university teaching hospital. On average, nearly 70 women attend antenatal care daily at this facility. The study involved pregnant women aged 18 years and above attending antenatal care at the facility. Pregnant women who exhibited severe mental health challenges limiting their ability to cooperate meaningfully or those who required intensive obstetric care were excluded from participation.

Informed consent was obtained through either written consent or verbal, recorded by the research assistants, depending on the participant's preference or literacy level. Study approval was obtained from the Muhimbili University of health and Allied Sciences (MUHAS) Institutional Ethical Review Board (No. DA.282/298/01.C/2029), MUHAS Senate Research and Publications Committee, the Executive Director of MNH, and the Head of the Department of Obstetrics and Gynecology at MNH. Participants were informed that those identified with early signs of psychological distress during the study would be offered access to mental health support. During the study, participants who screened as experiencing moderate to severe psychological distress were provided with psychoeducation and, where necessary, were referred to outpatient mental health services at MNH mental health department, in line with institutional and IRB ethical protocol.

### Sample size determination and sampling procedure

The study arrived at a minimum sample size of 287 participants, calculated using Cochran's formula $N = Z^2 P (1-P)/e^2$ based on a previous Ethiopian study reporting a prevalence of 21.5%, [14]. Other assumptions included a 5% margin of error, 95% confidence level, and a non-response rate of 10% of the required minimum sample size. Participants were selected using a probability sampling method, through a simple random sampling technique. To ensure randomness, participants were consecutively approached, as they arrived for their antenatal visits, and those who provided consent and drew 'in' from the lottery were recruited for the study.

### Data collection instruments and procedures

Data collection was done through face-to-face interviews with the principal investigator and two trained medical doctors serving as research assistants. To ensure consistency and quality of data collection, the research assistants went through a one-week intensive training covering the study objectives, ethical considerations, interview techniques, and the proper administration of study tools. The training emphasized standardized data collection procedures, maintaining participant confidentiality, and handling potential distress in participants sensitively. To enhance comprehension, the Swahili version of the questionnaire was administered, since it is the primary language spoken by participants. Additionally, a pre- test was conducted with 20 pregnant women attending antenatal care at the facility to assess the clarity, reliability, and cultural appropriateness of the questionnaire. Necessary adjustments were made based on the pre-test findings to improve the quality of data collection.

**Demographic information.** We collected data on social demographic characteristics, using a structured questionnaire which was developed by the researcher and informed by previous studies on maternal mental health within sub-Saharan Africa, ensuring contextual relevance to Tanzania [10,14,15]. Variables included, maternal age, marital status, level of education, occupation, all of which have been shown to influence mental health during pregnancy. Clinical characteristics such as parity, pregnancy intention, gestation age and history of previous pregnancy loss were also assessed.

**Maternal psychological distress.** We used Kessler Psychological Distress Scale (K-10); developed by Ronald C. Kessler and colleagues in 2002, a standard 10-item questionnaire designed to assess psychological distress as non-specific emotional states experienced by individuals over the previous four weeks [16]. The questions are rated on a 5-point Likert scale, with total scores ranging from 10 to 50, and it categorizes levels of distress as mild (20–24), moderate (25–29), and severe (30–50), with scores of 20 and above indicated psychological distress [16]. K-10 has demonstrated good reliability (Cronbach's α = 0.93), and in Tanzania, K10 has been validated and culturally adapted for use in clinical settings, showing good internal consistency and reliability (Cronbach's α = 0.86) [17]. This makes it a suitable and reliable tool for identifying psychological distress in the context of ANC, where resources for mental health assessment may be limited.

**Perceived social support.** We assessed perceived social support using the Oslo Social Support Scale (OSSS- 3) developed by Bøen, Dalgard, and Bjertness, a brief 3-item self-reported questionnaire, inquiring about the number of close confidants, level of interest and concern from others, and the quality of relationships with neighbors, scored within a four Likert scale [14]. This instrument evaluates perceived social support and classifies it as poor (3-8), moderate (9-11), good (12-14) [18]. OSSS=3 has been used in various African studies and demonstrates adequate contextual relevance, though a formal validation in Tanzania is yet to be established.

**Experience of intimate partner violence.** Intimate partner violence was assessed using a subset of questions from the Swahili version of WHO Multi-Country Study on Women's Health and Domestic Violence against Women tool developed in 2005, it has been validated and previously used by other studies in Tanzania [19–22]. The tool assesses three types of IPV: Psychological violence (4 items), which examined whether the participant's partner had insulted, humiliated, intimidated, or threatened them. Physical violence (6 items), which included questions about experiences such as slapping, hitting, pushing, kicking, choking, or being threatened with a weapon; and Sexual violence (1 item), which asked if the participant had been coerced into sexual intercourse against their will [23]. A woman was classified as having experienced IPV during pregnancy if she reported at least one instance of either psychological, physical, or sexual violence from her partner.

For the purpose of this study the assessment period was modified from the original tool's focus on IPV over the past 12 months to evaluating experiences specifically during the current pregnancy. This modification was made to improve the relevance of data to the antenatal care settings and to reduce recall-bias. While this tool adjustment tends to narrows the recall window from participants, previous studies suggest that IPV occurring during pregnancy can be reliably captured within the current pregnancy since it has significant health implication [22,23].

## Data processing and analyses

Data were reviewed for completeness by the principal investigator, followed by cleaning and coding to ensure consistency. The cleaned dataset was entered and analyzed using SPSS version 23. The outcome variable, maternal psychological distress was dichotomized based on a cutoff score of 20. Participants with scores of 20 or above were classified as having psychological distress, while those scoring below 20 were categorized as not having psychological distress. Crude and adjusted prevalence ratios (cPR and aPR) and their corresponding 95% confidence intervals (CIs) were estimated using Poisson regression model with robust variance to assess the relationship between maternal psychological distress and predictor variables. A p-value of $< 0.05$ was considered statistically significant.

## Results

### Sociodemographic and clinical characteristics of participants

All 287 eligible and consenting participants completed the study and were included in the final analysis. During data cleaning no participants were excluded due to missing or incomplete data. The mean age was 30.87 years (SD±4.3), with the

majority (74.1%) between 25 and 34 years old. Most participants were married (87.5%), more than one-third (35.9%) had attained a secondary education level and nearly three-quarters (74.2%) were employed.

At the time of enrollment, 61.3% of participants were in their third trimester, while 79.8% had given birth before. A notable proportion (26.5%) reported that their pregnancy was unintended, and 65.5% had experienced a previous pregnancy loss. Regarding social support, the majority (58.5%) reported moderate support, while 19.5% had poor perceived social support. Additionally, 42.2% of participants reported experiencing intimate partner violence (IPV) during pregnancy (Table 1)

## Prevalence of maternal psychological distress

Overall, 31.2% of pregnant women who were attending ANC reported to experience maternal psychological distress.

**Table 1. Distribution of socio-demographic and clinical characteristics of participants (n = 287).**

| Characteristics | N (%) |
|---|---|
| **Age (years)** | **Mean** 30.87, SD ± 4.3 |
| 18-24 | 16(5.6) |
| 25-34 | 213(74.1) |
| 35 and above | 58(20.3) |
| **Marital Status** | |
| In a relationship | 251(87.5) |
| Not in a relationship | 36(12.5) |
| **Education Level** | |
| Non-formal and Primary | 90 (31.4) |
| Secondary | 103(35.9) |
| Post-secondary | 94(32.7) |
| **Occupation** | |
| Employed | 213(74.2) |
| Unemployed | 74(25.8) |
| **Gestational age** | |
| First Trimester | 18(6.3) |
| Second Trimester | 93(32.4) |
| Third Trimester | 176(61.3) |
| **Parity** | |
| Nulliparous | 58(20.2) |
| Multiparous | 229(79.8) |
| **Pregnancy Intention** | |
| Intended | 211(73.5) |
| Unintended | 76(26.5) |
| **Loss of previous pregnancy** | |
| Yes | 188(65.5) |
| No | 99(34.5) |
| **Perceived Social support** | |
| Poor | 56(19.5) |
| Moderate | 168(58.5) |
| Good | 63(22) |
| **Experience of IPV** | |
| Yes | 121(42.2) |
| No | 166(57.8) |

## Factors associated with maternal psychological distress

The prevalence of maternal psychological distress was higher among younger participants, and those with moderate to poor social support. Younger women aged 18–24 years were significantly more likely to experience psychological distress compared to those aged 25 years and above (aPR = 1.64, 95% CI: 1.10–2.45, p = 0.016). Similarly, perceived social support showed a strong association with psychological distress, as women with poor support (aPR = 0.458, 95% CI: 0.321–0.652, p < 0.001) and those with moderate support (aPR = 0.325, 95% CI: 0.178–0.593, p < 0.001) were found to be at higher risk compared to women with good support. Participants with unintended pregnancies had a higher likelihood of distress in unadjusted analysis (cPR = 1.785, CI:1.288-2.474, p = 0.001), but the association was not statistically significant after adjusting for confounders (1.245(0.879-1.762) (Table 2).

## Discussion

This study aimed to assess the prevalence and associated factors of maternal psychological distress among pregnant women who were receiving antenatal care at Muhimbili National hospital in Dar-es-Salaam. It was hypothesized that maternal psychological distress is common during pregnancy and may be influenced by both individual and contextual factors.

Findings from this study revealed that 32.1% of pregnant women who were attending ANC at MNH experienced psychological distress. This prevalence is notably higher than the 19.1% reported in the general population in a study conducted in Mbeya and Songwe, Tanzania [24], and also higher than other African studies such as Ethiopia with a prevalence of 21.5% [14] and 26.5% in South Africa [25]. This discrepancy may be due to differences in study settings and population characteristics. The multi-center nature of the Ethiopian and primary health care settings in the South African study may have allowed for a broader representation of pregnant women. Our study was conducted in a single tertiary referral hospital, where women may be more likely to experience other medical challenges that could have contributed to feelings of distress, and women referred from lower-level healthcare facilities may already be experiencing anxiety and uncertainty about their pregnancy outcomes.

In our study, the younger maternal age and moderate to poor social support were significantly associated with maternal psychological distress. These findings are consistent with findings from Ethiopia [14] and Indonesia [26]. Particularly in Tanzania decreasing age was also found to increase the risk for overall maternal mental health morbidity during pregnancy [10] supporting its predisposition to psychological distress. A similar association between maternal age and mental health morbidity was also observed in high-income countries, like the United Kingdom, where through maternal self-report, women in their teens and 20s were found at greater risk of psychological distress during the antenatal period compared to mature pregnant women [27]. This could be attributed by an increased psychosocial demand during the preparation for and transitioning to motherhood and overwhelming nature of the responsibility and expectations associated with such role in the absence of an adequate support system which tend to exacerbate feelings of isolation and inadequacy [28]. Moreover, cultural and societal norms prevalent in Tanzania settings may also play a role, as younger single pregnant women may face stigma and rejection [29], which can add to their existing stressors and contribute to their experience of psychological distress. In contrast to this, a study in the United States [30] revealed younger age was not linked with maternal psychological distress, and this could be due to differences in social-cultural dynamics.

Pregnant women with moderate to poor social support were significantly more likely to experience psychological distress in our study. In line with this, a Tanzanian study looking at IPV among pregnant women revealed that, poor practical social support during pregnancy increased the risk of Intimate partner violence, which in turn contributed to the experiences of distress [22]. Along with that, the emotional and physical demands of pregnancy, combined with role transitions and competing responsibilities, can be overwhelming, and an inadequate support may impair coping mechanisms and overall wellbeing. There are evidences suggesting that women with strong family and social networks experience lower stress levels, and a lower likelihood of postpartum depression [31] and foster healthy development of unborn child [32].

**Table 2. Factors associated with maternal psychological distress, among pregnant women attending antenatal care clinic at Muhimbili National Hospital, Dar-es-salaam.**

| | Total | Reported Psychological Distress | cPR | P | aPR (95% CI) | Adj. P |
|---|---|---|---|---|---|---|
| **Maternal Age** | | | | | | |
| 18-24 | 16 | 11 (68.8) | 2.30(1.577-3.355) | 0.00 | 1.64(1.098-2.450) | 0.016 |
| 25+ | 271 | 81 (29.9) | Ref | | | |
| **Marital Status** | | | | | | |
| Married | 251 | 74(29.5) | Ref | | | |
| Not married | 36 | 18 (50.0) | 0.59(0.352-0.987) | 0.044 | 0.883(0.603-.1.294) | 0.524 |
| **Educational Level** | | | | | | |
| Primary or less | | 21(23.3) | 0.647(0.426-0.984) | 0.042 | 0.75(0.503-1.119) | 0.159 |
| Secondary or more | 197 | 71(36) | Ref | | | |
| **Occupation Status** | | | | | | |
| Working | 213 | 73 (34.3) | Ref | | | |
| Not working | 74 | 19(25.7) | 1.335(0.808-2.052) | 0.188 | 1.356(0.904-2.032) | 0.141 |
| **Parity** | | | | | | |
| Nulliparous | 58 | 17(29.3) | 0.895(0.576-1.391) | 0.622 | 0.759(0.489-1.179) | 0.220 |
| Multiparous | 229 | 75(32.8) | Ref | | | |
| **Gestational age** | | | | | | |
| First trimester | 18 | 7(38.9) | 1.141(0.617-2.109 | 0.674 | 1.262(0.677-2.354) | 0.464 |
| Second Trimester | 93 | 25(26.9) | 0.789(0.532-1.168) | 0.236 | 0.816(00.567-1.175) | 0.274 |
| Third Trimester | 176 | 60(34.1) | Ref | | | |
| **Pregnancy Intention** | | | | | | |
| Intended | 211 | 56(26.5) | Ref | | | |
| Unintended | 76 | 36(47.4) | 1.785(1.288-2.474) | 0.001 | 1.245(0.879-1.762) | 0.217 |
| **History of loss of previous Pregnancy** | | | | | | |
| No | 188 | 53(28.2) | Ref | | | |
| Yes | 99 | 39(39.4) | 0.716(0.512-1.00) | 0.050 | 0.828(0.529-1.296) | 0.410 |
| **Ever experience Intimate partner violence** | | | | | | |
| No | 166 | 52(31.3) | Ref | | | |
| Yes | 123 | 40(33.1) | 0.948(0.675-1.330) | 0.756 | 1.025(0.743-1.414) | 0.880 |
| **Perceived social support** | | | | | | |
| Poor | 56 | 38(67.9) | 0.377(0.275-0.517) | 0.000 | 0.458(0.321-0.652) | 0.00 |
| Moderate | 168 | 43(25.6) | 0.257(0.146-0.453) | 0.000 | 0.325(0.178-0.593) | 0.00 |
| Good | 63 | 11(17.5) | Ref | | | |

*Significant association (p-value <0.05), cPR-Crude prevalence ratio, aPR-Adjusted prevalence ratio

Although several studies have demonstrated a strong association between unintended pregnancy and psychological distress, our study did not find a statistically significant relationship. This may be partly explained by the timing of participant recruitment, as the majority of women in our sample were in their second or third trimesters. According to previous, meta-analyses of studies conducted in low and middle-income countries, report that women with unplanned pregnancy often delay initiating antenatal care [33,34], as a result, those experiencing high levels of distress earlier in pregnancy may not have been captured in our sample. Additionally, over time, some women may have adjusted psychologically to the pregnancy, thus reducing reported distress. It is also possible that marital status offered a protective effect, as the majority of women with unintended pregnancies in this study were married, this could have facilitated perceived support. Recent

research suggest that such support can reduce psychological distress during pregnancy, by buffering the emotional impact of pregnancy related stressors, including unintended pregnancy [35,36].

Regardless a substantial number of pregnant women in this study who reported an exposure to IPV, its association to maternal psychological distress was not found to be statistically significant. Partly could be explained by some of our sociocultural norms and systems, which tend to normalize some forms of violence like those related to psychological and sexual violence, resulting into women being more lenient, report them as less distressing, to avoid the stigma and shame that may be accompanied by their disclosure.

## Recommendations

To improve maternal mental health outcomes, this study emphasizes the importance of integrating routine mental health screening into antenatal care for early detection and timely intervention. Equipping healthcare providers with mental health awareness, effective communication, and basic counseling skills is essential for ensuring comprehensive antenatal care. Additionally, public health initiatives and policies should focus on raising awareness, strengthening social support systems, encouraging partner involvement, and promoting peer support networks to foster maternal well-being. Future studies could build on these results by exploring additional psychosocial and contextual factors affecting maternal mental health

## Limitations

While this study provides valuable insights into maternal psychological distress, certain limitations are worthy the acknowledgement. First, it was conducted in a single tertiary hospital, which primarily serves referred cases, limiting the generalizability of findings to the broader pregnant population in Dar es Salaam and Tanzania, especially those receiving care in primary healthcare settings. Women attending tertiary referral hospitals may have higher-risk pregnancies, potentially leading to an overestimation of psychological distress compared to the general population. Additionally, the study is subject to recall bias, particularly in responses to sensitive questions related to psychological distress and social experiences. Participants may have underreported or modified their responses due to stigma, social desirability, or embarrassment, affecting the accuracy of reported distress levels. Despite these limitations, this study provides critical primary data on maternal mental health and underscores the need for further research in diverse healthcare settings, including primary healthcare facilities and rural areas, to improve the applicability and generalizability of findings.

## Conclusion

This study reveals a high prevalence of maternal psychological distress among pregnant women who were attending antenatal care at Muhimbili National Hospital. Younger maternal age and poor social support were identified as significant risk factors. In Tanzania this study informs national policies aimed at improving maternal mental health, through early screening and targeted psychosocial interventions into routine antenatal care services. The findings may also guide similar effort in other low- and middle-income countries facing comparable maternal mental health challenges.

## Supporting information

**S1 File. Data set used in the analysis.**
(SAV)

## Acknowledgments

We sincerely thank Muhimbili National Hospital (MNH) and the Department of Psychiatry and Mental Health at Muhimbili University of Health and Allied Sciences for providing the platform to conduct this study. We are also grateful to the participants for their time and willingness to share their experiences, as well as to the healthcare providers and research assistants for their invaluable support during data collection.

## Author contributions

**Conceptualization:** Glory Cuthbert.

**Data curation:** Glory Cuthbert.

**Formal analysis:** Glory Cuthbert, Samuel Likindikoki.

**Methodology:** Glory Cuthbert.

**Resources:** Glory Cuthbert.

**Supervision:** Samuel Likindikoki, Deodatus Kakoko, Muzdalifat Abeid.

**Writing – original draft:** Glory Cuthbert.

**Writing – review & editing:** Samuel Likindikoki.

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
