## [Decision Letter · Decision Letter 0]

6 Jun 2025

PMEN-D-25-00141

Unveiling Maternal Psychological Distress: A Cross-Sectional Study among Pregnant Women Attending Antenatal Care at Muhimbili National Hospital, Dar-es-salaam, Tanzania.

PLOS Mental Health

Dear Dr. Cuthbert,

Thank you for submitting your manuscript to PLOS Mental Health. After careful consideration, we feel that it has merit but does not fully meet PLOS Mental Health’s publication criteria as it currently stands. Therefore, we invite you to submit a revised version of the manuscript that addresses the points raised during the review process.

EDITOR: Be sure to:

Critically address the changes raised by the reviewers

We look forward to receiving your revised manuscript.

Kind regards,

Kizito Omona, PhD

Academic Editor

PLOS Mental Health

Journal Requirements:

1. We noticed you have some minor occurrence of overlapping text with the following previous publication(s), which needs to be addressed:

- https://doi.org/10.21203/rs.3.rs-3287792/v1

- DOI:10.1186/1472-698X-13-4

- https://doi.org/10.3389/fpsyg.2021.627604

In your revision ensure you cite all your sources (including your own works), and quote or rephrase any duplicated text outside the methods section. Further consideration is dependent on these concerns being addressed.

Additional Editor Comments (if provided):

Critically address the changes raised by the reviewers

Reviewers' comments:

Reviewer's Responses to Questions

**Comments to the Author**

1. Does this manuscript meet PLOS Mental Health’s publication criteria?

Reviewer #1: Yes

Reviewer #2: Yes

2. Has the statistical analysis been performed appropriately and rigorously?

Reviewer #1: Yes

Reviewer #2: Yes

3. Have the authors made all data underlying the findings in their manuscript fully available (please refer to the Data Availability Statement at the start of the manuscript PDF file)?

Reviewer #1: Yes

Reviewer #2: Yes

4. Is the manuscript presented in an intelligible fashion and written in standard English?

Reviewer #1: Yes

Reviewer #2: Yes

Reviewer #1: Dear Authors

I have a few recommendations-

Abstract: You may remove the subheadings from abstract and put it into one single paragraph.

Introduction:

1st Paragraph: It is better to begin introduction with a generalized term rather than defining 'Psychological Distress' (PD). For example, I can suggest you to begin with terms like, Mental health/Emotional challenges during pregnancy. Then you can define PD. It will help the non-professional readers and students to understand the vast meaning of 'Psychological Distress'.

2nd Paragraph: Kindly include global prevalence of PD among women overall and then highlight the prevalence among pregnant women to differentiate the sensitivity of mental health at this period.

3rd Paragraph: Add information regarding contributing factors influencing distress among pregnant women in Tanzania to strengthen the justification. If there's no data in Tanzania related to this, you can look for data from neighboring or low- and middle- income countries around Tanzania so that the justification gets more highlight by mentioning the study gap from those countries.

Result:

Please mention the number of any missing or incomplete data while cleaning or how many participants dropped out of the study (if any). Otherwise, the result section is clearly defined with necessary tables.

I have been through the rest of the manuscript and I really appreciate the dedication of writing such a well-structured article along with rigorous statistical analysis.

Finally, PLOS encourages authors to reflect on why the findings matter, particularly for Tanzanians. You can briefly mention in the conclusion about potential policy relevance in Tanzania or similar LMIC contexts (1-2 sentences).

Wish you Good Luck

Reviewer #2: GENERAL COMMENT:

This is a relevant study on the prevalence and determinants of psychological distress among pregnant women seeking antenatal care at a tertiary hospital in Tanzania. The study is promising in terms of filling a gap in maternal mental health research in Tanzania. Particularly, exploring intimate partner violence as a determinant of maternal psychological distress is interesting. However, some revisions are required to make this paper ready for consumption.

MAJOR ISSUES:

1. While the results of the study are valid, the study's title does not seem to capture the fact that the study is specifically about the prevalence and determinants of psychological distress among pregnant women. It will be helpful to the audience to ensure that the title reflects these two concepts.

MINOR ISSUES:

1. Under 'methodology', specifically the paragraph on demographic information, it will be good to provide references to the studies that formed the basis for structuring the demography aspect of the questionnaire.

2. Under 'methodology', specifically section on experience of intimate partner violence, it will be helpful to provide references to studies in Tanzania that have validated and/or used the Women’s Health and

Domestic Violence against Women tool as you stated in line 4 of that section.

3. Still under the section on experience of intimate partner violence, can you explain whether reducing the the assessment period from 12 months to the current pregnancy (considering that for some women it might be as short as 3 months, etc.) does not significantly affect the tools validity? How sure are we of this?

4. In paragraph 5 of the discussion, you stated

"Looking at unintended pregnancy and its lack of association, a meta-analysis of studies conducted in low and middle-income countries, reports that women with unplanned pregnancy tend to delay beginning their antenatal care (26,27). While this cannot be surmised from the study findings, it is notable that majority of the participants were in their second and third trimesters, and that the study was unable to capture participants in their early stages of pregnancy."

Could you clarify how the findings of the studies you cited explain your finding of a lack of association between unintended pregnancy and psychological distress? It is not so clear from the paragraph.

**Do you want your identity to be public for this peer review?** For information about this choice, including consent withdrawal, please see our Privacy Policy

Reviewer #1: **Yes: ** Rehnuma Abdullah

Reviewer #2: **Yes: ** Nhyira Yaw Adjei-Banuah

---

## [Decision Letter · Decision Letter 1]

30 Jul 2025

PMEN-D-25-00141R1

Prevalence and Determinants of Maternal Psychological Stress: A Hospital-based Cross-sectional study, among Pregnant women attending Antenatal Care in Dar es Salaam, Tanzania

PLOS Mental Health

Dear Dr. Cuthbert,

Thank you for submitting your manuscript to PLOS Mental Health. After careful consideration, we feel that it has merit but does not fully meet PLOS Mental Health’s publication criteria as it currently stands. Therefore, we invite you to submit a revised version of the manuscript that addresses the points raised during the review process.

EDITOR: 

Address the comments raised by the two reviewers

We look forward to receiving your revised manuscript.

Kind regards,

Kizito Omona, PhD

Academic Editor

PLOS Mental Health

Journal Requirements:

Additional Editor Comments (if provided):

Address the comments raised by the reviewers

Reviewers' comments:

Reviewer's Responses to Questions

**Comments to the Author**

Reviewer #3: (No Response)

Reviewer #4: (No Response)

publication criteria?

Reviewer #3: Yes

Reviewer #4: Yes

3. Has the statistical analysis been performed appropriately and rigorously?

Reviewer #3: I don't know

Reviewer #4: Yes

4. Have the authors made all data underlying the findings in their manuscript fully available (please refer to the Data Availability Statement at the start of the manuscript PDF file)?

Reviewer #3: Yes

Reviewer #4: Yes

5. Is the manuscript presented in an intelligible fashion and written in standard English?

Reviewer #3: Yes

Reviewer #4: Yes

Reviewer #3: MAJOR ISSUES:

1- The research objectives should be stated equally in the abstract, introduction, and discussion.

Discussion

This study aimed to assess the prevalence of maternal psychological distress among pregnant women attending antenatal care.

introduction

To address this gap, this study aimed to assess the prevalence and associated factors of psychological distress among pregnant women attending antenatal care at Muhimbili National Hospital in Dar-es-Salaam, Tanzania

Abstract

This study aimed to assess the prevalence of maternal psychological distress and identifying key determinants among pregnant women receiving antenatal care (ANC) at the Muhimbili National Hospital (MNH) in Dar-es-Salaam.

The abstract states that the study of prevalence and factors, but the discussion only mentions the study of prevalence and does not specifically mention the statistical population.

MINOR ISSUES:

• Abstract

A. What do you mean by key factors?

This is vague. It would be better if key factors were specifically named.

• Introduction

A. All abbreviations (ANC…) should be written out in full the first time they appear in the text.

• Method

A. All abbreviations (MNH…) should be written out in full the first time they appear in the text.

B. Write details of the validity and reliability of the validated version of the Kessler Psychological Distress Scale (K-10) in Tanzania.

C. Please, write the details of the OSSS-3 and IPV tools.

Ex:

I. Year

II. Developers

III. Scoring method

IV. Validity and reliability of the original version

V. Validity and reliability of the Tanzanian version

Reviewer #4: The manuscript is relevant and well written. However, 2 issues have arisen:

1. What was the ethical approach for "disposing" the participants, if found with significant mental health issue?

2. what impact did you notice on the birth outcome ( ultimate outcome of interest)

**Do you want your identity to be public for this peer review?** For information about this choice, including consent withdrawal, please see our Privacy Policy

Reviewer #3: No

Reviewer #4: No

---

## [Editor Report · Decision Letter 2]

12 Aug 2025

PMEN-D-25-00141R2

Prevalence and Determinants of Maternal Psychological Stress: A Hospital-based Cross-sectional study, among Pregnant women attending Antenatal Care in Dar es Salaam, Tanzania

PLOS Mental Health

**Dear Dr. Cuthbert,**

Thank you for submitting your manuscript to PLOS Mental Health. After careful consideration, we feel that it has merit but does not fully meet PLOS Mental Health’s publication criteria as it currently stands. Therefore, we invite you to submit a revised version of the manuscript that addresses the points raised during the review process.

EDITOR: 

Odd Ratios are used for rare events (less than 10%). I can see that your prevalence is 31.2%. I advise that you use prevalence rate (PR) instead of Odd ratio (OR). So, COR will be replaced with cPR and AOR replaced with aPR.This minor revision can be be addressed in less than one week if you are not very busy

Please submit your revised manuscript by **26th August 2025.** If you will need more time than this to complete your revisions, please reply to this message or contact the journal office at mentalhealth@plos.org. Please include the following items when submitting your revised manuscript:

We look forward to receiving your revised manuscript.

Kind regards,

Kizito Omona, PhD

Academic Editor

PLOS Mental Health

Journal Requirements:

Additional Editor Comments (if provided):

Odd Ratios are used for rare events (less than 10%). I can see that your prevalence is 31.2%. I advise that you use prevalence rate (PR) instead of Odd ratio (OR). So, COR will be replaced with cPR and AOR replaced with aPR. This minor revision can be be addressed in less than one week if you are not very busy
---

## [Editor Report · Decision Letter 3]

29 Aug 2025

Prevalence and Determinants of Maternal Psychological Stress: A Hospital-based Cross-sectional study, among Pregnant women attending Antenatal Care in Dar es Salaam, Tanzania

PMEN-D-25-00141R3

**Dear Dr. Cuthbert,**

We are pleased to inform you that your manuscript 'Prevalence and Determinants of Maternal Psychological Stress: A Hospital-based Cross-sectional study, among Pregnant women attending Antenatal Care in Dar es Salaam, Tanzania' has been provisionally accepted for publication in PLOS Mental Health.

Best regards,

Kizito Omona [PhD]

Academic Editor

PLOS Mental Health

Thank your for addressing the changes